# Role of p-Coumaric Acid and Micronutrients in Sulfur Dioxide Tolerance in *Brettanomyces bruxellensis*

Mahesh Chandra [1,*], Patrícia Branco [1,2], Catarina Prista [1] and Manuel Malfeito-Ferreira [1]

1    Laboratório de Microbiologia, Linking Landscape Environment Agriculture and Food Research Center (LEAF), Instituto Superior de Agronomia, University of Lisbon, Tapada da Ajuda, 1349-017 Lisboa, Portugal; patricia.branco@ulusofona.pt (P.B.); cprista@isa.ulisboa.pt (C.P.); mmalfeito@isa.ulisboa.pt (M.M.-F.)
2    School of Engineering, Lusófona University, Campo Grande 376, 1749-024 Lisboa, Portugal
*    Correspondence: mchandra@isa.ulisboa.pt; Tel.: +351-21-365-3442

**Abstract:** Sulfite is a common preservative in wine, but the spoilage yeast *Brettanomyces bruxellensis* can produce volatile phenols even with the recommended sulfite dose. The purpose of this study was to examine how wine components, p-coumaric acid (a precursor of volatile phenols), and micronutrients influence culturability, viability, and volatile phenols production by *B. bruxellensis* under sulfite stress. In red wine, a high sulfite dose (potassium metabisulfite, 100 mg L$^{-1}$) led to an immediate death phase followed by growth recovery after two weeks. However, 4-ethylphenol (4-EP) was continuously produced by dead or nonculturable cells. Nonetheless, an event of growth recovery could not be observed in the case of the model wine. However, when the model wine was supplemented with minerals and vitamins, both growth recovery and 4-EP production were noticed, suggesting that the minerals and vitamins played an important role in maintaining the viability of cells under the sulfite stress. The yeast could also utilize the p-coumaric acid (p-CA) as an energy source, showing a specific growth rate of 0.0142 h$^{-1}$ with 1 mM of p-CA in model wine. Furthermore, the sulfite-stressed cells exhibited ATP production by means of proton efflux while utilizing the p-CA. This work highlights the novel finding that the conversion of p-CA into 4-EP provides sufficient energy for the cell to remain metabolically active under the sulfite stress.

**Keywords:** *Brettanomyces bruxellensis*; red wine spoilage; sulfur dioxide; sulfite stress; p-coumaric acid; VBNC; volatile phenols; 4-ethylphenol; wine micronutrients; vitamins

## 1. Introduction

The *Brettanomyces bruxellensis* is one of the major causes of red wine spoilage [1]. The yeast utilizes p- coumaric acid found naturally in grapes and produces the off-odor compound 4-ethylphenol (4-EP) that has a 'smoky', 'medicinal', or 'animal' (sensory threshold of 440 µg/L) aroma [2,3]. Unlike other wine yeast species, this yeast species can survive and proliferate long after alcoholic fermentation and remains active during storage and even after years of bottling by producing volatile phenols [2]. Sulfite or sulfur dioxide (SO$_2$) is added to the wine at different time intervals to stop the growth; using 20 to 30 mg L$^{-1}$ of free SO$_2$ is recommended for red wine ageing [4]. However, off-odors due to volatile phenol production by *B. bruxellensis* have been reported even after applying the recommended concentrations of sulfur dioxide [5,6].

In sulfite-treated wines, *B. bruxellensis* sometimes remains undiagnosed using routine laboratory culture techniques, which have been attributed to the induction of a viable but not culturable (VBNC) state [7,8]. The sulfite-induced VBNC (viable but nonculturable) state in *B. bruxellensis* exhibits a reduced glycolytic flux [9], suggesting that the glycolytic flux enzymes are not completely inhibited. However, in other yeast species, sulfur dioxide is known to inhibit key glycolytic flux enzymes such as glyceraldehyde-3-phosphate dehydrogenase (GAPDH), ATPase, alcohol dehydrogenase, aldehyde dehydrogenase, and



NAD$^+$-glutamate dehydrogenase. This causes the loss of adenosine triphosphate (ATP) generation and nicotinamide adenine dinucleotide phosphate (NADH) regeneration [10–12]. In *B. bruxellensis*, during the sulfite-induced VBNC state, the continuation of volatile phenols production indicates the functionality of hydroxycinnamic acid (HCA) converting enzymes (phenolic acid decarboxylase and vinylphenol reductase). This suggests a hypothesis that the conversion of HCA into volatile phenols facilitates yeast survival under sulfite stress. Whether it is, the cell metabolic state that influences volatile phenol production or the volatile phenols synthesis that confers better-surviving capabilities to the cells has remained unexplained so far.

The most relevant hydroxycinnamic acid (HCA) precursor of volatile phenols in wine is *p*-coumaric acid (*p*-CA) [13]. This molecule has been described as a preservative of natural foods [14] because of its antimicrobial activity [15,16]. However, *B. bruxellensis* is able to metabolize *p*-CA using a phenolic acid decarboxylase enzyme (PAD) that converts it into 4-vinylphenol (4-VP), which is further reduced to 4-EP using an NADH-dependent vinyl phenol reductase (VPR) [17,18]. It has been suggested that the decarboxylase action and the production of volatile phenols are related to yeast tolerance to toxic hydroxycinnamic acids [19,20]. Thus, it is conceivable that the yeast converts the toxic *p*-CA into the non-toxic 4-EP to survive. The conversion is supposed to provide energy to the cell to remain metabolically active since the conversion reaction yields ATPs. Therefore, it is worthwhile to investigate if the energy carrier ATPs are produced, even under sulfite stress, for the cell to remain active.

Major wine constituents like ethanol, sugar, and sulfur affect growth and volatile production by *B. bruxellensis* [21]. In addition, this species survives and produces volatile phenols under low residual sugar (<0.275 g L$^{-1}$) and yeast assimilable nitrogen (<6 mg N L$^{-1}$) that remain after primary fermentation [22–24]. However, the growth of this yeast may be reliant on pyridoxine, thiamine, and/or biotin, as evidenced by the inhibited growth observed when these vitamins are absent from the media [25–27]. Nevertheless, the impact of vitamins and minerals on the viability of *B. bruxellensis* and its production of 4-EP remains unexplored in our current understanding. Therefore, the aim of this study was to uncover the influence of minerals, vitamins, and *p*-CA on the behavior of sulfite-stressed *B. bruxellensis*, specifically focusing on its ability to produce 4-EP under wine conditions.

## 2. Materials and Methods

### 2.1. Yeast Strain and Maintenance

The yeast, *Brettanomyces bruxellensis* ISA 2211, was isolated from phenolic-tainted red wine in our laboratory [28] and maintained in slants of Glucose yeast peptone (GYP) medium (20 g L$^{-1}$ glucose, 5 g L$^{-1}$ yeast extract, 10 g L$^{-1}$ peptone, and 20 g L$^{-1}$ agar) at 4 °C.

Authentic red wines were acquired by blending multiple commercial red wines with <2.0 g L$^{-1}$ residual sugar. Sulfite concentrations were regulated using potassium metabisulfite or eliminated using acetaldehyde treatment [4]. The ethanol content was adjusted with a solution of 5.0 g L$^{-1}$ of tartaric acid (Merck, Darmstadt, Germany) or 99% pure ethanol (Merck, Darmstadt, Germany). The model wine medium was adapted from the K-medium [29] with some modifications. The medium contained glucose, 1.5 g L$^{-1}$ fructose, 1.5 g L$^{-1}$ glycerol, 10.0% *v*/*v* ethanol, 0.5 g L$^{-1}$ DL-malic acid, 0.01% *v*/*v* acetic acid, 4.0 g L$^{-1}$ L-lactic acid, 5.0 g L$^{-1}$ (NH$_4$)$_2$SO$_4$, 5.0 g L$^{-1}$ KH$_2$PO$_4$, 0.50 g L$^{-1}$ MgSO$_4$7H$_2$O, 0.132 g L$^{-1}$ CaCl$_2$ 2H$_2$O and 0.01 g L$^{-1}$ g p-Coumaric acid. The vitamin solution contained 0.01 g L$^{-1}$ biotin, 0.80 g L$^{-1}$ calcium pantothenate, 4.0 g L$^{-1}$ inositol, 1.6 g L$^{-1}$ niacin, 1.6 g L$^{-1}$ pyridoxin HCl and 1.6 g L$^{-1}$ thiamine HCl. The mineral solution contained 1.0 g L$^{-1}$ H$_3$BO$_3$, 0.2 g L$^{-1}$ KI, 0.4 g L$^{-1}$ Na$_2$MoO$_4$ 2H$_2$O, 0.08 g L$^{-1}$ CuSO$_4$5H$_2$O, 0.4 g L$^{-1}$ FeCl$_3$ 6H$_2$O, 0.8 g L$^{-1}$ MnSO$_4$ 4H$_2$O and 0.8 g L$^{-1}$ ZnSO$_4$ 7H$_2$O. The mineral and vitamin solutions were added at the concentration of 0.5 mL L$^{-1}$ to the model wine medium. The pH of both red wine and model wine medium was adjusted to 3.50 with NaOH (Merck, Darmstadt, Germany) or HCl (Merck, Darmstadt, Germany). Both wines

underwent sterilization by filtration using cellulose acetate membranes (0.22 μm pore size and 47 mm diameter, Millipore, Burlington, MA, USA) and were stored at 22 °C before yeast inoculation.

Throughout the incubation process, the samples were subjected to decimal dilution, and cellular culturability was assessed using surface plating 0.1 mL of the wine or model wine sample onto GYP medium in duplicate. Incubation was carried out at 25 °C for a duration of up to 7 days.

### 2.2. Culture Conditions

To prepare the *B. bruxellensis* ISA 2211 inoculum, the cells were grown in filter sterilized (0.22 μm pore size), 100 mL of Yeast Nitrogen Broth (6.7 g $L^{-1}$ YNB with 10.0% ethanol), with pH adjusted to $3.50 \pm 0.01$. The cells were incubated at 25 °C with orbital shaking at 120 rpm. Growth progress was monitored by measuring the absorbance at 640 nm. When the absorbance reached approximately 0.5 units, wines were inoculated with this inoculum to achieve an initial population of approximately $10^5$ cells $mL^{-1}$.

The inoculated wines were then placed in 100 mL Erlenmeyer flasks, capped with rubber plugs fitted with inserted hypodermic needles to provide minimal headspace. Throughout the incubation period, wine samples were decimally diluted, and cellular culturability was determined using surface-plating 0.10 mL of the samples onto the GYP medium in duplicates.

The specific growth rate (μc) was determined by calculating the slope of the growth curve during the exponential phase using the equation $x_t = x_0 + \mu_{ct}$, where $x_t$ and $x_0$ represent the biomass concentration or optical density (OD) at time t (hours) and t = 0, respectively. In all instances, the $R^2$ values of the curves were 0.996 or greater.

### 2.3. Sulfite Stress

The potassium metabisulfite (PMB) solution containing 57.60% of sulfur dioxide ($SO_2$) was used. The PMB concentrations of 25.0, 37.5, 50.0, 62.5, 75.0 and 100 mg $L^{-1}$ were used in both wine and model wine medium at pH 3.50 and 10% (*v/v*) ethanol.

### 2.4. $H^+$ Translocation

Cells were collected using centrifugation at $12,000 \times g$ for 3 min at 4 °C and then washed twice with ice-cold water. Suspensions were prepared in distilled water and kept on ice for a minimum of 1 h. To determine dry biomass, 0.10 mL of the cell suspension was desiccated using pre-weighed aluminum foil. The $H^+$ movements were measured by recording the pH of unbuffered cell suspensions in a 2.0 mL water-jacketed cell with magnetic stirring. The pH measurements were carried out using a standard pH meter (PHM62; Radiometer Copenhagen, Denmark) connected to a potentiometer recorder (BBC-Goerz Metrawatt SE460). To conduct $H^+$ efflux measurements, 0.10 mL of the cell suspension was combined with 0.80 mL of water. After adjusting the pH to 5.0, 100.0 μL of glucose was introduced into the solution, resulting in a final concentration of 0.1 M glucose. This triggered the $H^+$ extrusion process, leading to acidification in the unbuffered environment. The maximum rate of extracellular $H^+$ concentration increase, calibrated using 10 mM KOH, served as a measure of the $H^+$ extrusion activity [30].

### 2.5. Chemical Analysis

The 4-EP production was assessed following the procedure described by Bertrand [31]. In summary, volatile phenols were extracted from a 5.0 mL sample (with adjusted pH to 8 with NaOH (Merck)) using 1:1 (*v/v*) ether/hexane. The 4-EP was isolated by collecting the organic phase of the mixture resulting from three consecutive extractions. Quantification was accomplished using gas chromatography with 3,4-dimethylphenol serving as the internal standard. The gas chromatography analysis was performed using an Agilent 7820A GC-FID series with a FactorFour capillary column (ID 0.25 mm, length 15 m, film thickness 0.25 μm). The injector operated in splitless mode at 230 °C, and a 2 μL volume

was injected. The temperature of the detector was set to 250 °C. Hydrogen was used as the carrier gas at a flow rate of 0.1 mL min$^{-1}$. The oven temperature was initially set at 50 °C, increased to 215 °C at a rate of 10 °C min$^{-1}$, and finally raised to 250 °C at a rate of 20 °C min$^{-1}$.

## 3. Results

### 3.1. Effect of Sulfite on Cell Viability and 4-EP Production

Red wines were inoculated with 5 logs CFU mL$^{-1}$ initial population of *Brettanomyces bruxellensis* (*B. bruxellensis*) and added of potassium metabisulfite (PMB) concentrations ranging from 25 to 100 mg L$^{-1}$. A sharp death period, i.e., no culturability on surface media, was recorded under 75 and 100 mg L$^{-1}$ PMB (Figure 1). However, the cells regained culturability within 5 days of inoculation in the wines with 75 mg L$^{-1}$ PMB and 15 days with 100 mg L$^{-1}$ PMB. Interestingly, in the absence of culturable cells (<1 mL$^{-1}$), continuous 4-EP production was recorded, demonstrating the active metabolism of viable but nonculturable (VBNC) cells. However, the production of 4-EP by such cells was up to 69% less compared to the ones showing full culturability without sulfite stress (Figure 1).

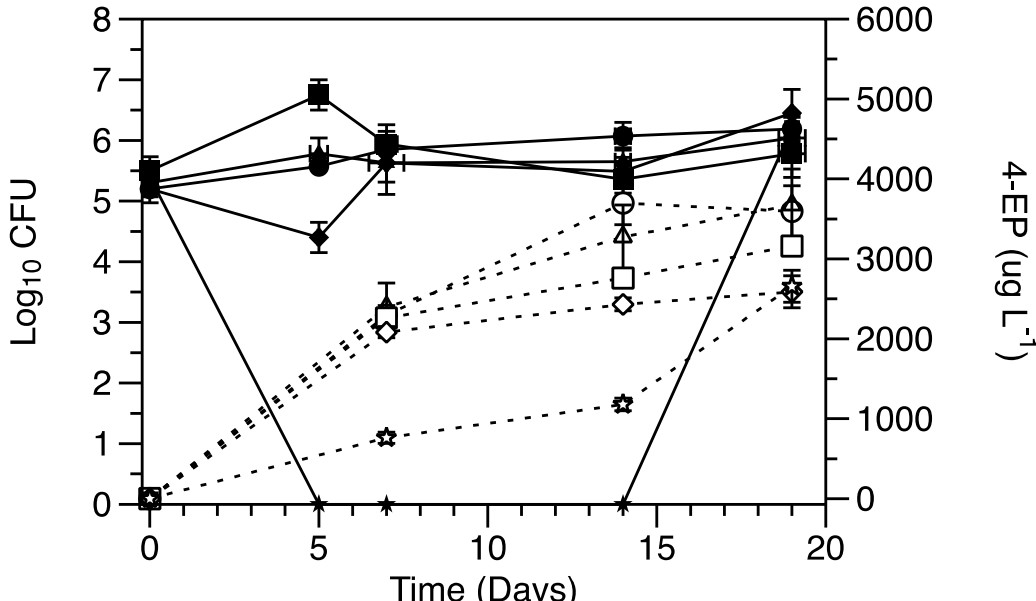

**Figure 1.** Growth of *B. bruxellensis* and the 4-ethylphenol (4-EP) production in real wine blends without the addition of potassium metabisulfite (PMB) (**circle**), with 25.00 mg L$^{-1}$ (**triangle**), 50.00 mg L$^{-1}$ (**square**), 75.00 mg L$^{-1}$ (**rectangle**) and 100.00 mg L$^{-1}$ (**star**) of PMB. The solid lines and filled symbols show the yeast growth in terms of CFU, while dotted lines and hollow symbols show 4-EP production.

### 3.2. Effect of Micronutrients on Cell Activity

In order to examine the effect of minerals and vitamins on growth and 4-EP production, the model wine medium adopted from K-medium was employed. Three different combinations were used: (1) model medium with vitamin solution (MV), (2) model medium with mineral solution (MM), and (3) model medium with vitamin and mineral solutions (MVM). The cell growth and 4-EP production were found to be much higher in the MVM compared to both the MV and MM medium. At 50.0 mg L$^{-1}$ PMB, the culturable cells were reduced to undetectable levels after the 6$^{th}$ day of inoculation in MM and MV medium. However, culturability was regained in the MV medium on the 10th day of inoculation. On the other hand, MM medium cells did not recover in any period of observations. Conversely, at the same PMB concentration (50.0 mg L$^{-1}$), no reduction in culturability was recorded in the

case of the MVM medium. The MVM medium cells always showed higher 4-EP production compared to MM and MV media, irrespective of the sulfite concentrations (Figure 2).

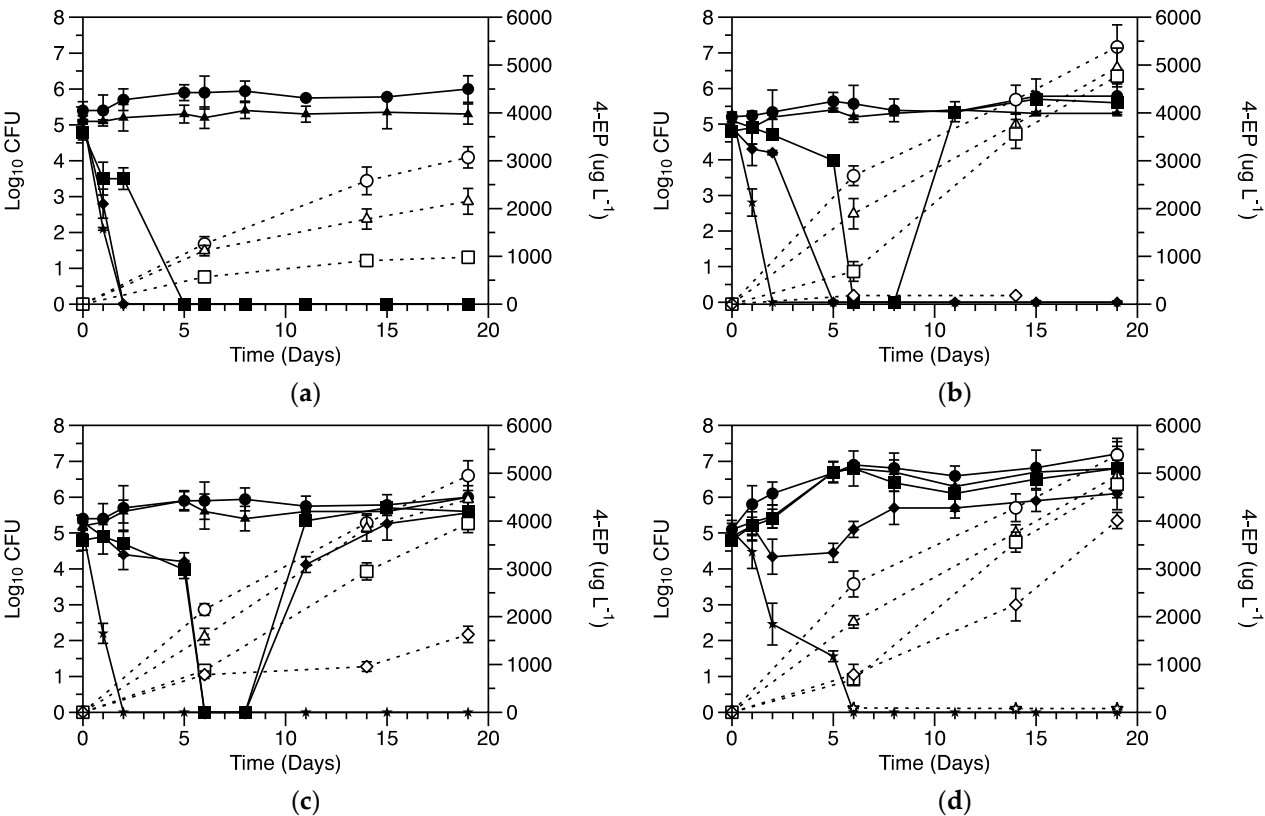

**Figure 2.** Growth of *B. bruxellensis* and the 4-ethylphenol (4-EP) production in model medium without minerals and vitamins (**a**), with minerals (**b**), with vitamins (**c**), and with minerals and vitamins (**d**) without any addition of potassium metabisulfite (PMB) (**circle**), with 25.00 mg L$^{-1}$ (**triangle**), 37.50 mg L$^{-1}$ (**square**), 50.00 mg L$^{-1}$ (**rectangle**), and 75.00 mg L$^{-1}$ (**star**) of PMB. The solid lines and filled symbols show the yeast growth in terms of CFU, while dotted lines and hollow symbols show 4-EP production.

### 3.3. Effect of p-Coumaric Acid on Growth and Proton Efflux

The concentration of 0.1 mM and 1 mM of p-CA in the model wine medium were used to investigate if *p*-CA supports yeast growth under sulfite stress. The *p*-CA was found to facilitate the growth of *B. bruxellensis,* exhibiting the specific growth rate 0.0142 h$^{-1}$ with 1.00 mM and 0.0134 h$^{-1}$ with 0.10 mM compared to 0.0171 h$^{-1}$ with 2.00 g L$^{-1}$ glucose when used as carbon source (Table 1).

**Table 1.** Growth rates of *B. bruxellensis* in the presence or absence of *p*-coumaric acid in model wine medium.

| Carbon Source | Specific Growth Rate (h$^{-1}$) | Doubling Time (h) |
| --- | --- | --- |
| Control | 0.0056 | 53.76 |
| Glu* | 0.0171 | 17.60 |
| Glu*+ 0.1 mM *p*-CA | 0.0282 | 10.67 |
| Glu* + 1.0 mM *p*-CA | 0.0271 | 11.11 |
| 0.1 mM *p*-CA | 0.0134 | 22.46 |
| 1.0 mM *p*-CA | 0.0142 | 21.20 |

*2 g L$^{-1}$ Glucose

The glucose uptake stimulates proton (H$^+$) efflux in yeast cells mediated by H$^+$-ATPase, providing cells with energy in the form of ATP. As expected, a higher plasma membrane H$^+$ efflux rate was reached for cells with glucose (0.026 mmol [g dry biomass]$^{-1}$ h$^{-1}$). Notably, this rate was similar to that of *p*-CA (0.022 mmol [g dry biomass]$^{-1}$ h$^{-1}$) (Figure 3). Assuming that efflux is the unique way for *B. bruxellensis* to cope with the sulfite stress, the assay with *S. cerevisiae* was conducted under similar conditions (Figure 4). Such efflux in the case of *S. cerevisiae* was not observed. On the other hand, while closely monitoring the path of efflux in *B. bruxellensis*, sulfite addition to the assay ceased the H$^+$ efflux in the beginning, which was restarted later and raised at a progressive rate (Figure 5). This suggests that the presence of p-CA enables the cells to synthesize ATPs under the stress of SO$_2$.

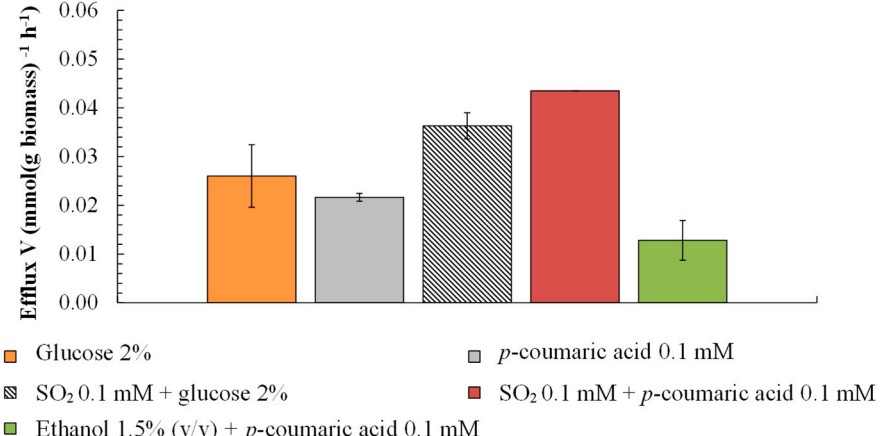

■ Glucose 2%  ■ *p*-coumaric acid 0.1 mM
▨ SO$_2$ 0.1 mM + glucose 2%  ■ SO$_2$ 0.1 mM + *p*-coumaric acid 0.1 mM
■ Ethanol 1.5% (v/v) + *p*-coumaric acid 0.1 mM

**Figure 3.** H$^+$ extrusion by *B. bruxellensis* in the presence of glucose, *p*-coumaric acid, ethanol, and SO$_2$. H$^+$ extrusion was measured by monitoring the pH of the medium with a pH electrode connected to a potentiometer recorder.

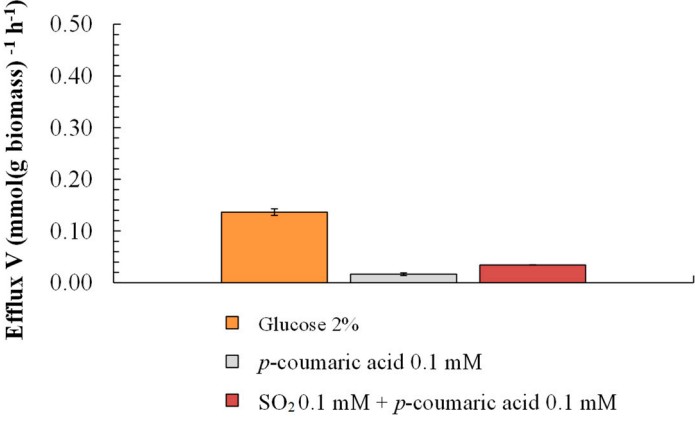

■ Glucose 2%
□ *p*-coumaric acid 0.1 mM
■ SO$_2$ 0.1 mM + *p*-coumaric acid 0.1 mM

**Figure 4.** H$^+$ extrusion by *S. cerevisiae* in the presence of glucose, *p*-coumaric acid, and SO$_2$. H$^+$ extrusion was measured by monitoring the pH of the medium with a pH electrode connected to a potentiometer recorder.

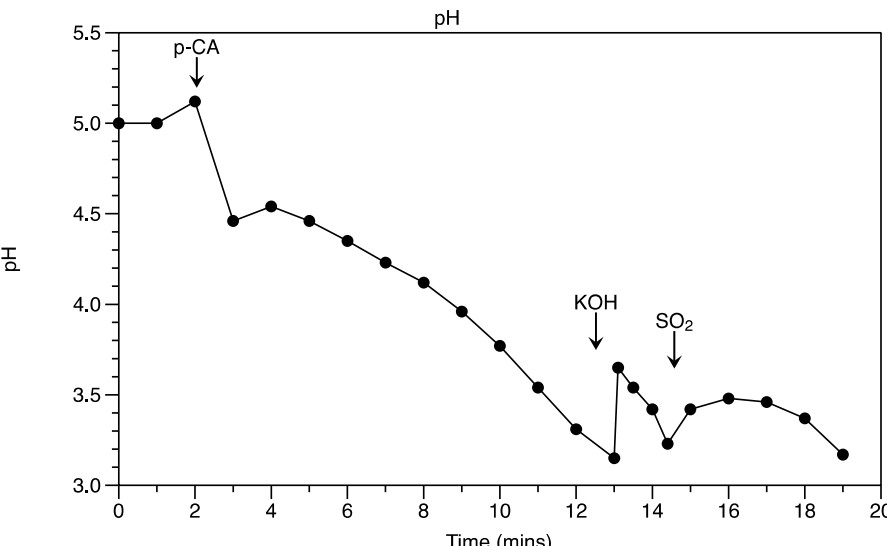

**Figure 5.** H$^+$ extrusion by *B. bruxellensis* in the presence of *p*-coumaric acid under SO$_2$ stress. H$^+$ extrusion was measured by monitoring the pH of the medium with a pH electrode connected to a potentiometer recorder.

## 4. Discussion

*B. bruxellensis* showed immediate death under the stress of SO$_2$ but regained culturability when observed at later periods, demonstrating the capability of the yeast to adapt to the SO$_2$ stress. The findings corroborate the results of others [7,9,32]. The recovery of cell culturability was dependent on the availability of SO$_2$ in the medium. The SO$_2$ level decreased in real wine over time, and recovery of growth took place when the SO$_2$ level lowered with the storage time [33]. However, Longin et al. [34] have speculated that SO$_2$ stress leads to the death of "sensitive cells" and that the remaining "resistant cells" can adapt themselves to new environmental conditions. The concept of "sensitive" and "resistant" cells seems interesting, but it can be meaningful when the population level is high. The sulfite-stressed cells in our experiments, showing the "death phase" or the VBNC state, could produce the volatile phenols (VPs) but far lower than that of control cells, indicating that the SO$_2$-triggered VBNC state cells can produce 4-ethylphenol but in a lower amount than that of culturable cells [9]. However, other authors did not find any production of volatile phenols by sulfite-stressed cells [32,33,35].

The survival of sulfite-stressed or VBNC state cells might also be dependent on the micronutrient's availability. It has been reported that removing important micronutrients such as trace vitamins like biotin can lead to the inhibition of *B. bruxellensis* growth [36]. The authors used the base medium that already contained both macronutrients and added different vitamins separately to study their role in the growth of *B. bruxellensis*. Our interest was to investigate if vitamins solely, or with the combination of minerals, support yeast survival under SO$_2$ stress and, thereby, 4-ethylphenol production. We found that cells showed better tolerance to SO$_2$ when the model medium was supplemented with a vitamin solution. The resistance was even better when the medium was supplied with vitamins and minerals; the cells regained growth after a short death period, even at a high sulfite dose (PMB, 50.00 mg L$^{-1}$). On the other hand, cells without vitamins and minerals did not recover growth, emphasizing the importance of both vitamins and minerals in the survival of yeast and volatile phenols production. These findings provide insights into the impact of vitamins and minerals on yeast behavior. However, the conclusions are limited by the use of a model wine medium, which does not accurately reflect the variability of mineral and vitamin content in actual wines due to variations in grape varieties [37,38]. Future research focusing on yeast behavior in wines with diverse vitamin and mineral profiles would be valuable.

We found that the precursor *p*-CA supported the growth of *B. bruxellensis* in sulfite-stress conditions, suggesting the role of *p*-CA in cell viability support. A similar observation on the growth of this yeast species has been made by other authors using synthetic wine medium or model wine culture medium [22,39,40]. This cinnamic acid improves the growth rate of *B. bruxellensis* and also significantly increases the activities of phenolic acid decarboxylase (PAD) activity and vinyl phenol reductase (VPR) enzymes [41] and thereby 4-EP production.

The continuous 4-EP production during the VBNC stage suggests that 4-EP production responsible enzymes, i.e., PAD and VPR, remain active in the cell during the VBNC state. It indicates that (i) decarboxylation of precursor hydroxycinnamic acids (HCAs) into vinyl phenol would enhance proton-motive force highly enough to derive ATP synthesis using F0F1-ATpase [42]; (ii) one mole of NADH would be generated per mole of cinnamic acid during conversion [8]. Therefore, it could be hypothesized that in the VBNC state, if 4-EP production occurs, the cell manages to synthesize ATP and regenerate NADH and thus maintains itself metabolically active. We worked on this hypothesis and observed that the cells that are under exposure to $SO_2$ could synthesize the ATPs in the presence of *p*-CA while assaying the plasma membrane ATPase activity, its action as a proton-pump and by measuring relative changes in plasma-membrane potential. According to past work, the mechanism of action of *p*-CA closely resembles that of weak acids, which can disturb intracellular pH and subsequently impact cell metabolism. [23]. In the presence of weak acids, such as ascorbic, benzoic, octanoic, succinic, or acetic acids, the cell employs a proton pump, plasma membrane $H^+$-ATPase (Pma1p) to reduce the concentration of protons and counteract the pH effects [22,24]. We have checked if $H^+$ efflux takes place in the presence of weak acids like benzoic and sorbic acids, but we could not notice any such efflux of $H^+$, suggesting that p-CA is being metabolized by the cell that results in activation of the proton pump. This pump has recently been reported in *B. bruxellensis* in response to *p*-CA in the growth media [43]. Later, these authors showed that *p*-CA induces the activation of transporters involved in the efflux of toxic compounds and drug resistance [44]. To this yeast, sulfite is a toxic compound, and it might be possible that proton efflux is one of the strategies to resist sulfite toxicity. A recent study shows that *B. bruxellensis* cells could manage to grow under the high-intensity light stress of 2500 and 4000 lux when the medium was supplied with *p*-CA. Contrarily, in the absence of *p*-CA, the yeast could not manage to grow under the same light-stress conditions [45]. This common hydroxycinnamic acid, *p*-CA, which is naturally present in wine, seems to play a significant role in supporting the cell to cope with chemical and physical stress and, therefore, deserves attention when developing strategies to control the *B. bruxellensis* in wine.

## 5. Conclusions

*B. bruxellensis* over competes with the other wine microbes due to its ability to survive in harsh conditions like low available residual sugar and high alcohol. It is interesting to know that minerals and vitamins play an important role in the survival of yeast, more importantly under sulfite stress. Future research should aim to pinpoint the specific vitamins or minerals that play a role in cell survival under sulfite exposure. It's worth highlighting that *p*-CA, a constituent of both grapes and wine, not only serves as a carbon source but also enables the cell to survive and produce 4-EP under sulfite stress. Our research marks the initial evidence suggesting that the conversion of *p*-CA into 4-EP supplies the necessary energy for the cell to maintain its metabolic activity when facing sulfite-induced stress. These findings serve as a foundation for future research exploring the role of *p*-CA in managing yeast survival under sulfite stress.

**Author Contributions:** Conceptualization, M.C. and M.M.-F.; methodology, M.C., P.B. and C.P.; formal analysis, M.C., C.P. and P.B.; writing—original draft preparation, M.C.; writing—review and editing, M.C. and M.M.-F. All authors have read and agreed to the published version of the manuscript.

**Funding:** This research was funded by Fundação para a Ciência e Tecnologia (FCT), Portugal, I.P., within DL 57/2016/CP1382/CT0012 to Mahesh Chandra, and strategic project UID/AGR/04129/2020 through Linking Landscape, Environment, Agriculture and Food Research Centre (LEAF).

**Data Availability Statement:** All data reported in this study are included in the manuscript.

**Acknowledgments:** The authors thank FCT, Portugal, for funding through DL 57/2016/CP1382/CT0012 to Mahesh Chandra and strategic project UID/AGR/04129/2020 (LEAF).

**Conflicts of Interest:** The authors declare no conflict of interest.

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
