# Peer review of "Role of p-Coumaric Acid and Micronutrients in Sulfur Dioxide Tolerance in Brettanomyces bruxellensis"

_beverages, doi:10.3390/beverages9030069_

Round 1
Reviewer 1 Report
The main item to correct is to specify how much free, bound, and molecular SO2 the two media contained. To understand the inhibition of yeast viability and activity it is important to know how much molecular SO2 was present.
Additional comments:
Line 82
.. commercial red wines 81 without residual sugar (< 2.0 g L-1).
Wines with less than 2 g/L of residual sugar still contain sugar. Thus, recommend to rewrite:
commercial red with less than 2.0 g L-1 residual sugar.
Line 87
Why was 5 g/L of malic acid and 4 g/L of lactic acid used in this medium? This is a rather unusual combination. With this much malic acid, wines usually do not contain this much lactic acid. The addition of 3 g/L acetic acid also makes this a very unusual medium, not very wine-like. Why were these extreme acid concentrations chosen?
Also, please specify whether L-malic acid or DL-malic acid was used.
Lines 119-121
What were the free and total SO2 concentrations in the test wines? Also, please calculate the molecular SO2 content in these wines.
The total amounts added do not tell us how much effective SO2 is present. You need to calculate the molecular SO2 content for these wines.
Lines 123-137
There is a duplication of text.
There are two Fig. 1. Please correct.
Line 334
B. bruxellensis over competes with the other wine microbes due to its ability to…
Suggest: B. bruxellensis over competes with the other wine microbes due to its ability to…
Reviewer 2 Report
The paper entitled: Role of p-Coumaric Acid and Micronutrients in Sulfur Dioxide Tolerance in Brettanomyces bruxellensis is very interesting especially for the wine industry.
Some changes can enhance the clarity and comprehension of this study's findings and conclusions.
Comments to the authors:
General: please check the whole document and make sure B. bruxellensis is spelled correctly (italics also needed)
Abstract: The novelty of the research is not very clear. Please clarify it along with the purpose of this study
Introduction: Very well addressed even to readers who are not familiar with the problem. I would like to suggest some changes:
1. A scheme of the second paragraph will be very helpful (and minimize some unnecessary details from the manuscript, lines 41-56).
2. You are descriping how sugar, nitrogen & ethanol affect B. bruxellensis growth. It is more preferable to change/add any literature about the nutrients since their affect you are interested in (lines69-75).
Materials and Methods: 2.5 Chemical analysis:
1. How did you achieve pH adjustment?
2. please add the ratio of ether-hexane
3. what was the extraction procedure? Be more specific.
Results: As your study is in wine model, have you consider that different grape varieties can affect the outcome? Add some preliminary comments about that if possible.
Conclusions: Fortify the future research according to your results (lines339-340.
Good luck!
Round 2
Reviewer 2 Report
You've made all the necessary changes in order for your manuscript to be even more interested and clarified in the insights it provides. It can be accepted at its currant form.